# Influenza and Respiratory Syncytial Virus Infections in Pediatric Patients during the COVID-19 Pandemic: A Single-Center Experience

**DOI:** 10.3390/children10010126

**Published:** 2023-01-07

**Authors:** Aušra Steponavičienė, Sigita Burokienė, Inga Ivaškevičienė, Indrė Stacevičienė, Daiva Vaičiūnienė, Augustina Jankauskienė

**Affiliations:** Clinic of Children’s Diseases, Faculty of Medicine, Institute of Clinical Medicine, Vilnius University, LT-03101 Vilnius, Lithuania

**Keywords:** COVID-19, SARS-CoV-2, RSV, influenza A/B, coinfections, children, screening, pediatric emergency department

## Abstract

The overlap of coronavirus disease 2019 (COVID-19) with other common respiratory pathogens may complicate the course of the disease and prognosis. The aim of the study was to evaluate the rates, characteristics, and outcomes of pediatric patients with severe acute respiratory syndrome coronavirus 2 (SARS-CoV-2), respiratory syncytial virus (RSV), influenza A/B infections, and their coinfections. A single-center prospective cross-sectional study was performed at the pediatric emergency department in Vilnius from 1 October 2021 to 30 April 2022. In total, 5127 children were screened for SARS-CoV-2, RSV, and influenza A/B. SARS-CoV-2 PCR tests were positive for 21.0% of children (1074/5127). The coinfection rate of respiratory viruses (RSV, influenza A) in patients with COVID-19 was 7.2% (77/1074). Among the 4053 SARS-CoV-2 negative patients, RSV was diagnosed in 405 (10.0%) patients and influenza A/B in 827 (20.4%) patients. Patients with COVID-19 and coinfection did not have a more severe clinical course than those with RSV or influenza infection alone. RSV and SARS-CoV-2 primarily affected younger patients (up to 2 years), while the influenza was more common in older children (4–10 years). Patients infected with RSV were more severely ill, reflected by higher hospitalization proportion and need for respiratory support.

## 1. Introduction

Seasonal epidemics of respiratory syncytial virus (RSV) and influenza viruses cause significant morbidity and mortality among young children worldwide [1,2]. In the cold season, acute respiratory infection symptoms trigger a significant proportion of primary care visits and hospitalizations [3,4]. The highest numbers of RSV hospitalizations were reported in children <1 year of age, which tend to peak in the winter months [5]. Every year, seasonal influenza viruses are estimated to cause >100 million illnesses and 870,000 hospitalizations globally for acute lower respiratory infection among children aged under 5 years [6,7].

Over the course of the coronavirus disease 2019 (COVID-19) pandemic, the epidemiology of these viruses changed dramatically. Initially, RSV and influenza A/B were low in circulation in 2020 [8]. Globally-induced restrictive measures such as social distancing, lockdowns, and mask wearing, limited the transmission of all respiratory viruses [9]. During the 2021/2022 cold season, cases of RSV and influenza increased and their characteristics changed, although published data are as of yet unavailable regarding both the increased numbers and changed aspects of these infections [8,10]. Countries are reporting incidence peaks at different times, in contrast to the trends of previous years before the COVID-19 pandemic [8,11]. The extent to which the COVID-19 pandemic and mitigation efforts have impacted the circulation of common respiratory viruses and potential interactions among them remains unclear. The overlap of COVID-19 with other respiratory pathogens may complicate the course of disease and prognosis [12]. Coinfections with COVID-19 emerge as a new concern.

The aim of the study was to evaluate the rates, characteristics, and outcomes of patients with severe acute respiratory syndrome coronavirus 2 (SARS-CoV-2) infection, RSV infection, influenza A/B, and their coinfections in a single Lithuanian center.

## 2. Materials and Methods

A single-center prospective cross-sectional study was performed during the usual influenza season, from 1 October 2021 to 30 April 2022. Patients were recruited at Vilnius University Hospital Santaros Klinikos, designated as a major hospital for admissions of pediatric patients with COVID-19 in the northeast region of Lithuania. The study was approved by the Vilnius Regional Biomedical Research Ethics Committee (No. 2020/8-1269-737). This article is a continuation of the research started earlier; two articles were written on the topic of COVID-19: PMID: 34805046, https://doi.org/10.3389/fped.2021.749641 (accessed on 27 December 2022).

Patients under 18 years who had acute respiratory symptoms and were tested for COVID-19 and other respiratory viruses (influenza, RSV) in the pediatric emergency department (PED) were enrolled in the study. Nasopharyngeal swabs were taken and real-time reverse-transcriptase PCR tests were performed for SARS-CoV-2, influenza A/B viruses, and RSV by Xpert Xpress CoV-2/Flu/RSV plus, Cepheid with 100% specificity and 100% sensitivity.

Basic characteristics (age, gender), clinical symptoms of respiratory distress (including shortness of breath and increased work of breathing), blood oxygen level (SpO2, %), the need for supplemental oxygen (SpO2 less than 92% for more than 6 h) and outcome data (hospitalized/discharged from PED, duration of hospitalization, admissions to PICU, recovered/deceased) were obtained from patient electronic medical records. The supplemental oxygen was provided by oxygen masks and nasal cannulas. The clinical data correspond to the first hours of admission time.

Patients were stratified into five groups (A–E) according to SARS-CoV-2, RSV, and influenza A/B test positivity: Group A: SARS-CoV-2 test positive, other respiratory virus tests negative; Group B: SARS-CoV-2 test positive, influenza A/B test positive; Group C: SARS-CoV-2 test positive, RSV test positive; Group D: only RSV test positive; Group E: only influenza A/B test positive (Figure 1).

Statistical analyses were performed with R software (version 4.1.0, R Foundation for Statistical Computing, Vienna, Austria). Categorical data were presented as frequencies and percentages and analyzed using Pearson’s chi-square or Fisher’s exact test where appropriate. For continuous data, medians/interquartile range (IQR) were calculated, and the Kruskal–Wallis rank sum test was used to compare groups. Pairwise comparisons between groups were performed using the Pearson’s chi-square, Fisher’s exact test exact binomial test, or Mann–Whitney U test, as appropriate. The Bonferroni adjustment method was used to adjust for multiple testing. The segmented regression was used to assess changes in trends in the number of cases by months within disease groups using the R package segmented [13]. An initial analysis was performed to determine the best number of change points using the Bayesian Information Criterion as the selection criterion. Five separate segmented regression models with the selected number of change points were fitted, and 95% confidence intervals (Cis) were calculated for the segmented regression line and estimated change points. A *p* < 0.05 is considered as statistically significant.

## 3. Results

### 3.1. Demographical Characteristics

The number of pediatric emergency department visits was 19,532. In total, 5127 children were screened for SARS-CoV-2 and other viruses (RSV and influenza A/B) at the PED. SARS-CoV-2 PCR tests were positive for 21.0% of children (1074/5127). The coinfection rate of respiratory viruses (RSV, influenza A) in patients with COVID-19 was 7.2% (77/1074): 2.2% with RSV, and 4.9% with influenza A. Influenza B was not detected. Among the 4053 SARS-CoV-2 negative patients, only RSV was diagnosed in 405 (10.0%) patients, and only influenza A/B in 827 (20.4%) patients. Influenza A was detected in 820 (99.2%) of these patients, and influenza B was detected in 7 (0.8%) patients (Figure 1).

The characteristics of Groups A–E are shown in Table 1. The distribution by gender was similar in all groups; the distribution by age differed significantly (Figure 2). The youngest patients fell into the COVID-19 and RSV groups (A, C, and D), with the majority of those aged 0–2 years (673, 67.5%; 20, 83.3%; and 288, 71.1%, respectively). The majority of patients with influenza (Groups B and E) were aged 4–10 years (31, 58.5%, and 472, 57.1%, respectively). The median age of patients with RSV infection and only COVID-19 infection was significantly lower than in patients with influenza (*p* < 0.001).

### 3.2. Clinical Features

Respiratory distress was more frequent in patients with RSV infection (Group C 37.5% and group D 18%, *p* < 0.001). The lowest saturation level was also in Groups C and D (20.8% and 18.3%, respectively, *p* < 0.001), compared to other groups. In the group with only SARS-CoV-2 (Group A), supplemental oxygen was required in 1.5% (15) of cases, whereas in the groups with only RSV infection or in RSV with COVID-19, this percentage was much higher: 17.8% in Group D and 12.5% in Group C (*p* < 0.001). None of the patients were intubated.

### 3.3. Seasonal Differences

The infection monthly distribution was significantly different among most groups, especially in Groups A, D and E: in Group A the majority of patients were diagnosed in February, Group B—in March, Group C—in January, Group D—in December, and Group E—in April (*p* < 0.001) (Figure 3).

The initial analyses carried out to determine the number of change points found that all segmented regression models have one change point. The best segmented regression fit was found in Group E with a change point at the fifth month (February 2022) (adjusted R-squared = 0.9098, change point estimate 5, 95% CI 3.2–6.8); Group D with a change point at the third month (December 2021) (adjusted R-squared = 0.7761, change point estimate 2.9, 95% CI 0.8–4.9); and Group A with a change point between the fifth and sixth month (February 2022 and March 2022) (adjusted R-squared = 0.7192, change point estimate 5.6, 95% CI 4.1–7.1). However, in Groups B and C, the 95% confidence intervals for the estimated change points were wide and covered the entire range of months analyzed (Figure 4).

### 3.4. The Proportion of Hospitalization

The proportion of hospitalization was highest in patients with only RSV infection (Group D, 34.6%) compared with other groups (*p* < 0.001). The lowest hospitalization was in patients with influenza A/B (Group B—3.8%, Group E—6.2%). The longest hospital stay (5 days) was recorded in Group D. Significant differences in the average duration of hospitalization were found among Groups A and D, and Groups D and E, *p* < 0.001.

Patients with a single COVID-19 and a single RSV infection had higher admission to PICU rates (2.9% in group A and 3.0% in Group D), as compared to influenza A/B groups (*p* = 0.002).

### 3.5. Outcome

One patient, a 5-year-old, died. He was SARS-CoV-2 PCR positive, and had myelodysplastic syndrome as an underlying disease. All other hospitalized patients were discharged if their condition improved, and there was no fever or need for supplemental oxygen.

## 4. Discussion

Our study compares relevant data about COVID-19 coinfection in children. We tested over 5000 children of different age groups with respiratory symptoms for SARS-CoV-2 and other viruses (RSV and influenza A/B) during the 2021/2022 cold season at a tertiary hospital. SARS-CoV-2 PCR tests were positive for 21.0% of children, illustrating that nearly one-fifth of children visiting due to respiratory symptoms had them caused by SARS-CoV-2. The second most common pathogen was the influenza virus, which was responsible for another one-fifth of the cases. Despite the high numbers of both infections registered in February and March, coinfections with these pathogens were uncommon, accounting for 5% of cases. The seasonality of RSV differed from influenza, and SARS-CoV-2 and the peak were registered in December. This may explain why RSV coinfections with SARS-CoV-2 were even less common. In general, the coinfection rate of respiratory viruses (RSV, influenza A) in patients with COVID-19 was 7.2%. Other countries reported similar results. In Bulgaria, from November 2020 to mid-March 2022, 242 SARS-CoV-2 positive patients (children and adults) were tested for seasonal respiratory viruses, and 24 (9.9%) cases of coinfections were detected [14].

According to the national statistics for Lithuania, during the 2020/2021 season, only 302 laboratory confirmed influenza cases were registered in the country, and none were hospitalized [15]. As compared to pre-pandemic seasons when the incidence of influenza was high, it appeared that influenza disappeared in 2020/2021. Data from our earlier study contribute to these findings, as only eight cases (out of all 83 tested due to respiratory symptoms) were positive for influenza in 2020/2021, and no cases of coinfection with SARS-CoV-2 were detected [16]. As soon as COVID-19 infection control measures were relaxed, the influenza cases re-emerged. During the period of our study, all of the most commonly identified viruses (influenza A, B, and RSV) were again present among our patients.

The vast majority of patients in COVID-19 and RSV groups were aged 0–2 years, whereas influenza was more common in children aged 3–11 years. It is known that RSV affects young children and presents in a complicated way [1], while influenza is more common in older children [17]. Since the beginning of the COVID-19 pandemic, children accounted for a small proportion of all cases, but these numbers had a tendency to increase. Children aged 5 years and older were most commonly affected in the pediatric group [18]. The situation dramatically changed when the Omicron variant became predominant. Disease rates in children are now the highest they have been since the start of the pandemic [19,20,21]. Data are not available by age groups in Lithuania, but numbers in other countries show that cases in children 0–4 years were the highest, especially in regard to COVID-19 hospitalizations [21,22]. Our findings contribute to these data, showing that children up to 2 years of age, not eligible for vaccination, were the most vulnerable to COVID-19.

Reported data in a systematic review and meta-analysis show no significant association between influenza coinfection and need for supplemental oxygen or deaths among COVID-19 patients. No significant association was found between RSV coinfection and deaths among COVID-19 patients [23]. Similar findings are reported in our study: patients with COVID-19 and coinfections did not have clinically more severe disease, as the hospitalization proportion and the need for supplemental oxygen were similar or lower compared with RSV or influenza infections alone. SARS-CoV-2 positive patients with a coinfection were not more frequently hospitalized compared with those who had no coinfection. Patients infected with RSV were more severely ill (especially compared with COVID-19), as the hospitalization proportion and need for respiratory support were the highest.

Since the start of the COVID-19 pandemic, according to CDC data, there were 1544 deaths in the pediatric population, while in adults there were over 100,000 deaths [24]. Data from the UNICEF registry show that there were more deaths from COVID-19 among children (under 20 years of age) at over 16,000 deaths [25]. A respiratory phenotype, which is more common in younger children, was associated with higher mortality. Mortality in pediatric populations is strongly associated with comorbidities [26]. According to country statistics, in Lithuania there were 6 child deaths from COVID-19 since the beginning of the pandemic, while in adults there were 9413 deaths [27]. In our study, all children underwent short-term hospitalization, with one death.

The strength of our study is the high number of variously aged children tested for SARS-CoV-2 and other viruses with clinical data. This study adds value due to the significant number of patients who presented at the PED due to viral infection, which might otherwise be controlled by vaccination. These results may help to improve health care strategies in Lithuania. The limitation of this study is missing data on underlying diseases, patient vaccination status against influenza infection and COVID-19. The statistical analysis was complicated because groups B and C were smaller compared with A, D, and E groups. We compared the influenza and RSV activities only, while other human respiratory viruses were not considered.

## 5. Conclusions

During the 2021/2022 season, three viral respiratory tract infections (SARS-CoV-2, influenza, and RSV) with clinically significant impact were recorded in pediatric patients. The frequency of coinfection was low. COVID-19 and RSV infections were more common in patients under the age of 2 years, while influenza affected older children (3–11 years). Patients infected with RSV were more severely ill compared to those with COVID-19, as shown in the proportion of hospitalizations and need for respiratory support. Patients infected with COVID-19 and RSV or influenza did not present with a more complicated clinical course, as their frequency of hospitalization, need for supplemental oxygen, or admission to PICU was similar or lower as compared to patients with single infections.

## Figures and Tables

**Figure 1 children-10-00126-f001:**
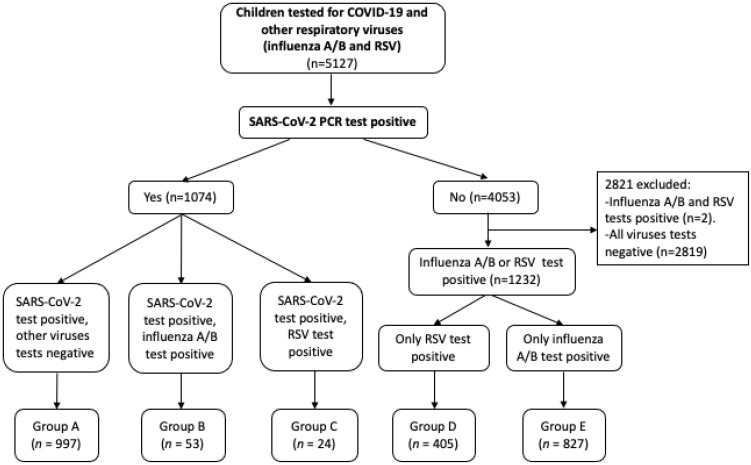
Stratification of enrolled patients. COVID-19—coronavirus disease 2019, PCR—polymerase chain reaction, RSV—respiratory syncytial virus, SARS-CoV-2—severe acute respiratory syndrome coronavirus 2.

**Figure 2 children-10-00126-f002:**
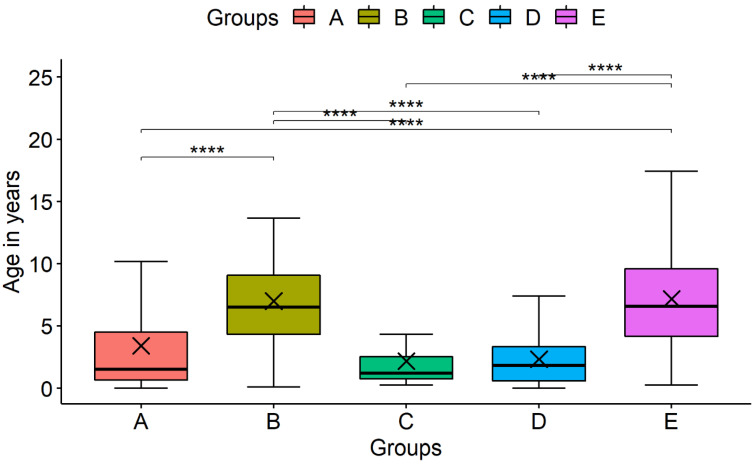
Comparison of age distribution in Groups A–E. **** *p* < 0.001, except between A and C; A and D; B and E; C and D groups. Group A: SARS-CoV-2; Group B: SARS-CoV-2 and influenza A/B; Group C: SARS-CoV-2 and RSV; Group D: RSV; Group E: influenza A/B.

**Figure 3 children-10-00126-f003:**
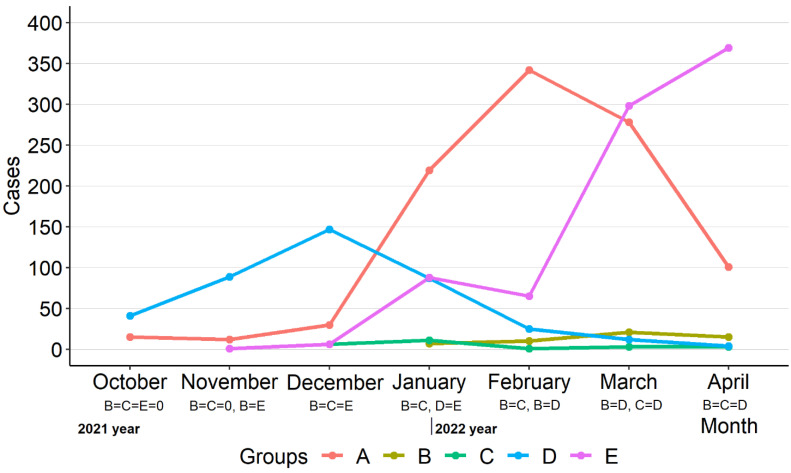
Distribution of cases (Groups A–E) monthly. Group A: SARS-CoV-2; Group B: SARS-CoV-2 and influenza A/B; Group C: SARS-CoV-2 and RSV; Group D: RSV; Group E: influenza A/B. The equality symbol indicates statistically insignificant differences between the proportions of the groups, and also indicates when the proportions of the groups being compared were zero, in which case the statistical test was not applied (B, C and E in October 2021, B and C in November 2021).

**Figure 4 children-10-00126-f004:**
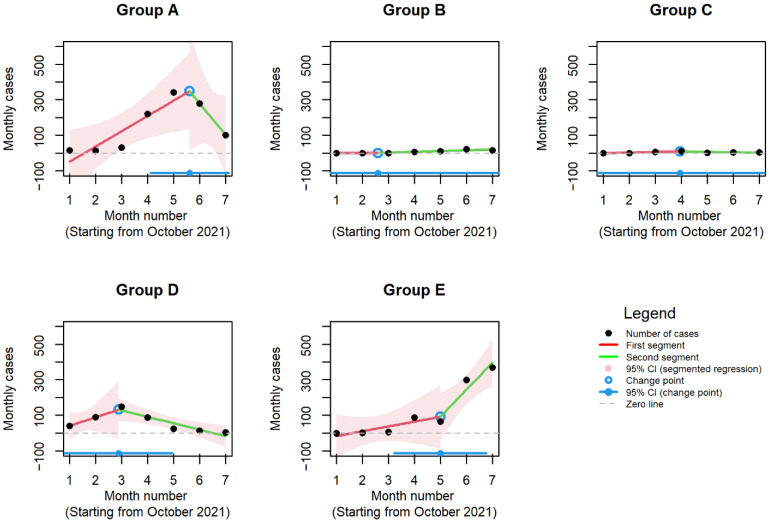
Results of segmented regression (Groups A–E). Group A: SARS-CoV-2; Group B: SARS-CoV-2 and influenza A/B; Group C: SARS-CoV-2 and RSV; Group D: RSV; Group E: influenza A/B.

**Table 1 children-10-00126-t001:** Characteristics of Groups A–E.

Characteristics	A gr. Only SARS-CoV-2 Test Positive	B gr. SARS-CoV-2 and Influenza A Tests Positive	C gr. SARS-CoV-2 and RSV Tests Positive	D gr. Only RSV Test Positive	E gr. Only Influenza A/B Test Positive	*p*-Value	Post Hoc ***
Total no. (F, %)	997 (44.2)	53 (43.4)	24 (41.7)	405 (44.0)	827 (45.1)		
Median age, years (IQR)	1.5 (0.7–4.5)	6.5 (4.3–9.1)	1.2 (0.8–2.5)	1.8 (0.6–3.3)	6.6 (4.2–9.6)	<0.001	1, 4, 5, 6, 9, 10
Respiratory distress * no.(%)	69 (6.9)	1 (1.9)	9 (37.5)	73 (18.0)	11 (1.3)	<0.001	2, 3, 4, 5, 6, 9, 10
SpO2 < 95% no.(%)	33 (3.3)	1 (1.9)	5 (20.8)	74 (18.3)	16 (1.9)	<0.001	2, 3, 6, 9, 10
The need for supplemental oxygen	15 (1.5)	0	3 (12.5)	72 (17.8)	3 (0.4)	<0.001	3, 6, 9, 10
Hospitalization no.(%)	170 (17.1)	2 (3.8)	5 (20.8)	140 (34.6)	51 (6.2)	<0.001	3, 4, 6, 10
Average duration of hospitalization, days—median (IQR)	3.0 (2.0–4.0)	2.5 (2.3–2.8)	4.0 (3.0–4.0)	4.0 (3.0–5.3)	3.0 (2.0–4.0)	<0.001	3, 10
Admission to PICU ** no.(%)	29 (2.9)	1 (1.9)	0	12 (3.0)	5 (0.6)	0.002	4, 10

* Respiratory distress includes shortness of breath and increased work of breathing. ** PICU—Pediatric Intensive Care Unit. F—female. *** Pairwise comparisons between groups with Bonferroni adjustment were performed. Statistically significant differences between groups are coded as follows: 1—A and B, 2—A and C, 3—A and D, 4—A and E, 5—B and C, 6—B and D, 7—B and E, 8—C and D, 9—C and E, 10—D and E.

## Data Availability

Not applicable.

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
