# Peer review of "Influenza and Respiratory Syncytial Virus Infections in Pediatric Patients during the COVID-19 Pandemic: A Single-Center Experience"

_children, 2023, doi:10.3390/children10010126_

Round 1

Reviewer 1 Report

The authors presented a relevant experience of a hospital focused on identifying the rates, characteristics, and outcome of patients with severe acute respiratory syndrome coronavirus 2 (SARS-CoV-2) infection, RSV infection, influenza A/B and 15 their coinfections in Vilnius, Lithuania.

Some changes should be done before considering the manuscript as a potential work for publication.

First, considering that the work was limited to one institution in Lithuania, I suggest including it in the title.

Second, the abstract should be written in the format established in the author's guidelines.

Third, please provide the sensitivity and specificity of the lab test used to diagnose COVID-19, RSV and other viruses. Also, please consider reporting the COVID-19 lineages identified during the study. 

Results

Please clarify the information from table 1. Does the clinical data correspond to the admission time? (i.e. SpO2 <95% no). 

Does "the need for supplemental oxygen" includes nasal cannula, ventury and orotracheal intubation?

Please check if the information provided in line 110 is correct "The majority of patients with influenza (Groups B and E) were age 3-11 years (36, 67.9% and 585, 70.8%, respectively)". According to table 1, the IQR age for groups B and E are 6.5 (4.3 - 9.1) and 6.6 (4.2-9.6).

Ages in table 1 and the text are presented in years, but the age in figure 2 are in months, please, be consistent. Also, I recommend improving the quality of the image according to the journal guidelines. 

Lines 128 and 131 are quite relevant. Nevertheless, the presented data is too descriptive. Please, consider an additional analysis to compare the trends such as a joint point regression analysis  https://www.nature.com/articles/s41598-019-40806-0 ; https://www.researchgate.net/publication/350976629_Did_domestic_travel_restrictions_slow_down_the_COVID-19_pandemic_in_Saudi_Arabia_A_joinpoint_regression_analysis

In lines 144 and 148 you have estimated a hospitalization rate. Please include a description of how did you get to the rate in the methods section. It was a rate or a proportion? Please clarify.

Line 152 you have described the case of an inpatient with comorbidity (myelodysplastic syndrome). There were more inpatients with other comorbidities? How did you deal with it in the analysis? Please include a column in table 1 with the comorbidities(listing it and the frequencies).

Also, if it is possible to add the COVID-19 and influenza vaccination status of the participants?

Please, check the limitation section according to the type of design, the timeline and the analysis that you have performed. I have some trouble understanding why the "patient vaccination status against influenza and COVID-19 infection" is a limitation of your study.

Regards,

Author Response

1. First, considering that the work was limited to one institution in Lithuania, I suggest including it in the title.

Thank you for the suggestion. The study was conducted in one big university centre in Lithuania, though the results do not reflect the situation in Lithuania as a whole. To avoid a long and complicated title, we will add this information to the methods and abstract.

2. Second, the abstract should be written in the format established in the author's guidelines.

We adjusted the number of words in the abstract and changed the format.

3. Third, please provide the sensitivity and specificity of the lab test used to diagnose COVID-19, RSV and other viruses. Also, please consider reporting the COVID-19 lineages identified during the study

We supplemented the lab tests with specificity and sensitivity data. We have not identified virus subtypes.

Sensitivity and specificity of tests in table.

Diagnostic kit name 

Analyte 

Specificity  

Sensitivity 

Sensitivity 

Xpert Xpress CoV-2, Cepheid 

SARS-CoV-2 RNR 

100 % 

100 % 

0.0200 PFU/mL 

Xpert Xpress CoV-2/Flu/RSV plus, Cepheid 

SARS-CoV-2 RNR 

100 % 

100 % 

0.2-4 TCID50/ml /100-412 copies/ml for different types of SARS-CoV-2 strains

Influenza A RNR 

100 % 

100 % 

0.0159-100 TCID50/ml for different types of influenza A strains and < 1 pg/µl for Avian influenza A 

Influenza B RNR 

100 % 

100 % 

0.025-60 TCID50/ml for different types of influenza B strains

RSV RNR 

100 % 

100 % 

0.1-0.5 TCID50/ml for different types of RSV strains

Xpert Xpress Flu/RSV, Cepheid 

Influenza A RNR 

100 % 

100 % 

0.013-0.750 TCID50/ml for different types of influenza A strains 

Influenza B RNR 

100 % 

100 % 

0.19-0.4 TCID50/ml for different types of influenza B strains

RSV RNR 

100 % 

100 % 

0.79-2.3 TCID50/ml for different types of RSV strains

TaqPath COVID-19 CE-IVD RT-PCR Kit, Applied Biosystem 

SARS-CoV-2 RNR 

100 % 

100 % 

10 SARS-CoV-2 GCE/reaction 

Results

4. Please clarify the information from table 1. Does the clinical data correspond to the admission time? (i.e. SpO2 <95% no). 

Yes, the clinical data corresponds to the admission time.

5. Does "the need for supplemental oxygen" includes nasal cannula, ventury and orotracheal intubation?.

Unfortunately, we do not have such data.

6. Please check if the information provided in line 110 is correct "The majority of patients with influenza (Groups B and E) were age 3-11 years (36, 67.9% and 585, 70.8%, respectively)". According to table 1, the IQR age for groups B and E are 6.5 (4.3 - 9.1) and 6.6 (4.2-9.6).

Thank you for the comment. We changed the age group from 3-11 years to 4-10 years.

7. Ages in table 1 and the text are presented in years, but the age in figure 2 are in months, please, be consistent. Also, I recommend improving the quality of the image according to the journal guidelines. 

We improved the quality of the image Figure 2 and changed the age from months to years.

8. Lines 128 and 131 are quite relevant. Nevertheless, the presented data is too descriptive. Please, consider an additional analysis to compare the trends such as a joint point regression analysis  https://www.nature.com/articles/s41598-019-40806-0 ; https://www.researchgate.net/publication/350976629_Did_domestic_travel_restrictions_slow_down_the_COVID-19_pandemic_in_Saudi_Arabia_A_joinpoint_regression_analysis

We did an additional analysis.

9. In lines 144 and 148 you have estimated a hospitalization rate. Please include a description of how did you get to the rate in the methods section. It was a rate or a proportion? Please clarify.

Thank you for the note. We meant the proportion, not the frequency.

10. Line 152 you have described the case of an inpatient with comorbidity (myelodysplastic syndrome). There were more inpatients with other comorbidities? How did you deal with it in the analysis? Please include a column in table 1 with the comorbidities(listing it and the frequencies).

Unfortunately, the comorbidities of all patients were not in the scope of our analysis. As only one patient died, we described him in more detail.

11. Also, if it is possible to add the COVID-19 and influenza vaccination status of the participants?

Unfortunately, such data was not collected We meant that this is the limitation of our study.

12. Please, check the limitation section according to the type of design, the timeline and the analysis that you have performed. I have some trouble understanding why the "patient vaccination status against influenza and COVID-19 infection" is a limitation of your study.

We corrected the sentence.    

Reviewer 2 Report

The authors produced a well-conducted study with significant content with the report of the seasonal spread and clinical features of SARS-CoV-2 and other respiratory viruses (RSV and Influenza) mono- and co-infections among Lithuanian children.

My impression of the manuscript is overall favorable. The design of the study is appropriate, and the relevance of the topic is high.

However, I suggest some changes for consideration for publication, as follows. Moreover, in most of the manuscript’s sections, the English grammatical structure of the sentences and the English vocabulary are amendable. Therefore, a full-text review by a native English speaker is mandatory for consideration for publication. 

Introduction: The introduction is adequate. I would suggest completing the last sentence (lines 50-52) by adding the setting of the study.

Methods: The methods section is well-organized and clear.

Results: The results section appears difficult to read in all its parts, mainly due to the high rate of the same word’s repetitions and the absence of a clear distinction between the different outcomes of the study.

To improve the readability of the paragraph, I would suggest showing the results in different sub-paragraphs (eg. Demographical characteristics of the population study, clinical features, seasonal differences in the infections’ peak, and hospitalization rate), describing any changes among the different infection groups.

Discussion: The discussion section’s contents are relevant, and the Authors provided a good overview of the current scientific evidence on the topic, comparing their findings with previous studies. However, it appears difficult to read, due to the lack of grammar structure and poor English vocabulary in many parts. As stated above, a full-text review by a native English speaker is mandatory for consideration for publication.

Author Response

1. However, I suggest some changes for consideration for publication, as follows. Moreover, in most of the manuscript’s sections, the English grammatical structure of the sentences and the English vocabulary are amendable. Therefore, a full-text review by a native English speaker is mandatory for consideration for publication.

A native American speaker reviewed the text. We can provide contact information.

2. Introduction: The introduction is adequate. I would suggest completing the last sentence (lines 50-52) by adding the setting of the study.

Thank you for the suggestion. We added the setting of the study.

3. Results: The results section appears difficult to read in all its parts, mainly due to the high rate of the same word’s repetitions and the absence of a clear distinction between the different outcomes of the study. To improve the readability of the paragraph, I would suggest showing the results in different sub-paragraphs (eg. Demographical characteristics of the population study, clinical features, seasonal differences in the infections’ peak, and hospitalization rate), describing any changes among the different infection groups.

We added sub-paragraphs.

4. Discussion: The discussion section’s contents are relevant, and the Authors provided a good overview of the current scientific evidence on the topic, comparing their findings with previous studies. However, it appears difficult to read, due to the lack of grammar structure and poor English vocabulary in many parts. As stated above, a full-text review by a native English speaker is mandatory for consideration for publication.

A native American speaker reviewed the text. We can provide contact information.

Reviewer 3 Report

I read with interest the manuscript of A. Steponavičienė et al, "Influenza and respiratory syncytial virus infections in pediatric patients during the COVID-19 pandemic". The epidemiology of respiratory viruses during the pandemic is of great interest.

The manuscript requires a number of changes to improve its quality. Below are my comments:

Introduction

Line 32: It should be noted that especially in cold seasons, acute respiratory infections put significant pressure on tertiary systems (hospitals) through increased hospitalisations and Emergency Department presentations. For example, you can use reference PMID: 32517775 to support such a statement.

Lines 40-44: An example of how SARS-CoV-2 and influenza circulation varied in the 2021/22 season can be found and cited in this article on data from Romania: PMID: 35626363.

Lines 46-49: I don't think you need to make this hypothesis yourself. In April 2022, Swets et al (PMID: 35344735) published a report in the Lancet on the impact of SARS-CoV-2 - influenza - RSV co-infections. I suggest that you rephrase your idea and integrate this article into your manuscript.

Methods

Line 55: Use "influenza", not "flu", it is the more academic term for articles.

The methods part is insufficiently described, is my major comment, and requires explanation, as you state that you conducted a retrospective and prospective study.

Describe what each of these stages entailed:

- Period studied

- Patient selection criteria (how were retrospective patients chosen, but prospective?)

Why did you choose to conduct a retrospective and prospective analysis?

Why the Ethics Committee approval is from 2020?

How many pediatric patients in total were in your hospital during this period? It is important to see the size of your data in relation to referrals to the hospital.

Did you perform multiplex RT-PCR, or did you test separately for each of the 3 viruses reported? Cw kits did you use? Absolutely all patients included in the study were confirmed by RT-PCR for any of the 3 viruses. Here it is very important to explain how you selected patients for testing. Because there is a difference between testing and including patients in a prospective study and taking data from a hospital's computer system about previously tested patients randomly without any clear selection criteria. Your data may suffer from selection bias. Please pay particular attention to the detail of the methods.

Results

Results start abruptly with reference to Table 1. Start the results by presenting demographic data of the whole studied group and then move on to the analysis of the studied groups.

Figures 2 and 3 should be redone with avoidance of capital letters and with more careful graphics. Pay attention also to the resolution of the figures, please improve the resolution.

Statistical analysis is simplistic. I would have liked to see more of a multivariate analysis with highlighting dependencies between risk factors in children both in terms of aetiology (mono/co-infections, virus type) and the particularities of children (age, sex, days since onset, chronic conditions). This is my second major comment.

Discussion

Line 157 - I disagree with this statement. Perhaps in terms of local analysis of data in your country, but globally there have been a number of reports published on viral circulation in pandemics and the impact on the pediatric population.

Lines 184-185: "It is known that RSV generally affects young children, and influenza is more common in older children [13]." - I disagree with this statement, which is why I tried to access reference number 13, which does not have all the data needed to be identified - valid link, date accessed, web page name. Indeed RSV causes forms of disease with a much greater clinical impact among infants, but that does not mean that the rest of the children are not equally affected. From clinical experience, we often tend (perhaps also for economics) to test for RSV especially infants and children under 2 years of age, hence the misconception that RSV infects young children. But at the same time since multiplex RT-PCR tests have been used it has been observed that RSV affects children of all ages equally well as adults. At the same time, influenza is again a non-age-invariant virus, and children under 5 years of age are a risk factor for severe disease. A reference you can use to document the impact of influenza in children and RSV co-infections can be found here: PMID: 34767579.

All in all, the discussions are ok, it would just deserve a review after reviewing the results.

Conclusions

Line 225 - I would not use the term most common but say viral infections with "clinically significant impact", because even rhinoviruses for example are much more common than influenza, RSV and SARS-CoV-2 combined.

The frequency of co-infections was low because a number of factors contributed here, worth mentioning in the discussion section to support this conclusion.

References

Please be careful when citing references, especially websites.

***Please find attached the iThenticate similarity check and address any potential issues.   

Author Response

The manuscript requires a number of changes to improve its quality. Below are my comments:

Introduction

  1. Line 32: It should be noted that especially in cold seasons, acute respiratory infections put significant pressure on tertiary systems (hospitals) through increased hospitalisations and Emergency Department presentations. For example, you can use reference PMID: 32517775 to support such a statement.

Thank you for your suggestion. We added additional information and the article.

  1. Lines 40-44: An example of how SARS-CoV-2 and influenza circulation varied in the 2021/22 season can be found and cited in this article on data from Romania: PMID: 35626363.

Thank you for your suggestion. We added the article.

  1. Lines 46-49: I don't think you need to make this hypothesis yourself. In April 2022, Swets et al (PMID: 35344735) published a report in the Lancet on the impact of SARS-CoV-2 - influenza - RSV co-infections. I suggest that you rephrase your idea and integrate this article into your manuscript.

Thank you for your suggestion. We cited the article.

Methods

  1. Line 55: Use "influenza", not "flu", it is the more academic term for articles. 

Thank you for the note. We changed the word.

The methods part is insufficiently described, is my major comment, and requires explanation, as you state that you conducted a retrospective and prospective study. 

Describe what each of these stages entailed:

  1. Period studied: From October 1, 2021, to April 30, 2022
  2. Patient selection criteria (how were retrospective patients chosen, but prospective?):

We included patients prospectively after Ethics Committee approval: patients under 18 years who had acute respiratory symptoms and were tested for COVID-19 and other respiratory viruses (influenza, RSV) in the pediatric emergency department (PED) were enrolled in the study.

  1. Why did you choose to conduct a retrospective and prospective analysis?

We included patients prospectively after Ethics Committee approval and then analysed the data.

  1. Why the Ethics Committee approval is from 2020?

This article is a continuation of the research started earlier; two articles were written on topic COVID-19: PMID: 34805046, https://doi.org/10.3389/fped.2021.749641.

  1. How many pediatric patients in total were in your hospital during this period? It is important to see the size of your data in relation to referrals to the hospital.

We added additional data.

  1. Did you perform multiplex RT-PCR, or did you test separately for each of the 3 viruses reported? Cw kits did you use? Absolutely all patients included in the study were confirmed by RT-PCR for any of the 3 viruses. Here it is very important to explain how you selected patients for testing. Because there is a difference between testing and including patients in a prospective study and taking data from a hospital's computer system about previously tested patients randomly without any clear selection criteria. Your data may suffer from selection bias. Please pay particular attention to the detail of the methods.

Nasopharyngeal swabs were taken, and real-time reverse-transcriptase PCR tests were performed for SARS-CoV-2, influenza A/B viruses and RSV by Xpert Xpress CoV-2/Flu/RSV plus, Cepheid with 100 % specificity and sensitivity. Patients under 18 years who had acute respiratory symptoms and were tested for COVID-19 and other respiratory viruses (influenza, RSV) in the pediatric emergency department (PED) were enrolled in the study.

Results

  1. Results start abruptly with reference to Table 1. Start the results by presenting demographic data of the whole studied group and then move on to the analysis of the studied groups. 

Thank you for your suggestion, we made corrections accordingly.

  1. Figures 2 and 3 should be redone with avoidance of capital letters and with more careful graphics. Pay attention also to the resolution of the figures, please improve the resolution.

We improved the quality of images Figure 2 and Figure 3.

  1. Statistical analysis is simplistic. I would have liked to see more of a multivariate analysis with highlighting dependencies between risk factors in children both in terms of aetiology (mono/co-infections, virus type) and the particularities of children (age, sex, days since onset, chronic conditions). This is my second major comment.

We made an additional statistical analysis of seasonal differences. The types of viruses were not analysed. A statistical analysis was performed on age and gender. Unfortunately, the comorbidities of all patients were not in the scope of analysis. As only one patient died, we described him in more detail. We added it to the limitations.

Discussion

  1. Line 157 - I disagree with this statement. Perhaps in terms of local analysis of data in your country, but globally there have been a number of reports published on viral circulation in pandemics and the impact on the pediatric population. 

We made corrections.

  1. Lines 184-185: "It is known that RSV generally affects young children, and influenza is more common in older children [13]." - I disagree with this statement, which is why I tried to access reference number 13, which does not have all the data needed to be identified - valid link, date accessed, web page name. Indeed RSV causes forms of disease with a much greater clinical impact among infants, but that does not mean that the rest of the children are not equally affected. From clinical experience, we often tend (perhaps also for economics) to test for RSV especially infants and children under 2 years of age, hence the misconception that RSV infects young children. But at the same time since multiplex RT-PCR tests have been used it has been observed that RSV affects children of all ages equally well as adults. At the same time, influenza is again a non-age-invariant virus, and children under 5 years of age are a risk factor for severe disease. A reference you can use to document the impact of influenza in children and RSV co-infections can be found here: PMID: 34767579.

Thank you for your valuable comments. We meant that young children more often have RSV complications, citation was added.

All in all, the discussions are ok, it would just deserve a review after reviewing the results.

Conclusions

  1. Line 225 - I would not use the term most common but say viral infections with "clinically significant impact", because even rhinoviruses for example are much more common than influenza, RSV and SARS-CoV-2 combined. 

Thank you for the note, we made corrections.

Round 2

Reviewer 1 Report

The authors presented a second version of the manuscript.

I have suggested changing the title. I must insist on including the place and location due to the fact the research was limited to one Hospital.

Second, considering that it was a prospective design, it is hard to understand why you don´t have access to clinical data such as the type of supplemental oxygen requirements and the comorbidities. These aspects are quite relevant considering the type of analysis you have performed.

Finally, I respectfully invite you to review the limitations paragraph. Please consider the timeline, the type of design, the missing information and data,  and that groups B and C are were smaller compared with A, D and E groups. 

In addition, I would like to invite you to consider in performing a multivariate analysis to evaluate the independency between sociodemographic and clinical features. 

Regards,

Author Response

  1. I have suggested changing the title. I must insist on including the place and location due to the fact the research was limited to one Hospital.

We changed the title: „Influenza and respiratory syncytial virus infections in pediatric patients during the COVID‐19 pandemic: a single-center experience“.

  1. Second, considering that it was a prospective design, it is hard to understand why you don´t have access to clinical data such as the type of supplemental oxygen requirements and the comorbidities. These aspects are quite relevant considering the type of analysis you have performed.

The data is from emergency department forms, the information about comorbidities isn‘t included.

The supplemental oxygen was provided by oxygen masks and nasal cannulas. None of the patients was intubated.

  1. Finally, I respectfully invite you to review the limitations paragraph. Please consider the timeline, the type of design, the missing information and data,  and that groups B and C are were smaller compared with A, D and E groups. 

We added this information to limitations paragraph.

  1. In addition, I would like to invite you to consider in performing a multivariate analysis to evaluate the independency between sociodemographic and clinical features. 

Unfortunately, we don‘t have information about sociodemographic information.

Reviewer 3 Report

The authors improved the manuscript following the first round of revision. I still consider that the figures need to be corrected with the avoidance of capital letters. Please use lower case. 

Author Response

The authors improved the manuscript following the first round of revision. I still consider that the figures need to be corrected with the avoidance of capital letters. Please use lower case. 

Thank you for your note. We changed the figures.
